# Antioxidant Activity of the *Prunus mahaleb* Seed Oil Extracts Using *n*-Hexane and Petroleum Ether Solvents: In Silico and In Vitro Studies

Zhawen Noori Hussein [1,*], Hoshyar Abdullah Azeez [2] and Twana Salih [1]

[1] Department of Pharmacognosy & Pharmaceutical Chemistry, College of Pharmacy, University of Sulaimani, Sulaymaniyah 46001, Iraq; twana.salih@univsul.edu.iq
[2] Department of Medical Laboratory, College of Health and Medical Technology, Sulaimani Polytechnic University, Sulaymaniyah 46001, Iraq; hoshyar.azeez@spu.edu.iq
* Correspondence: zhawen.hussein@univsul.edu.iq; Tel.: +964-(0)-770-5059955

**Abstract:** *Prunus mahaleb* L., also known as white mahaleb, and native to the Kurdistan region of Iraq, has significant nutraceutical and therapeutic ingredients. The seeds are rich in conjugated fatty acids with small quantities of cyanogenic glycosides, coumarin derivatives, and flavonoids. The contents of the seeds were extracted with the Soxhlet apparatus using *n*-hexane and petroleum ether solvents, separately. Gas chromatography-mass spectrometry (GC-MS) was used to recognize the chemical composition of the compounds. The radical scavenging activity was performed for the total extracts from *n*-hexane and petroleum ether solvents using 2,2-diphenyl-1 picrylhydrazyl (DPPH) assay and compared with quercetin as a positive control. Furthermore, molecular docking was performed for the identified compounds against five enzymes that have main roles in intracellular oxidation. Afterwards, drug-like properties and bioactivity predictions were applied for all compounds using Molinspiration software. The results showed four phthalate derivatives, six saturated fatty acids (SFAs), five monounsaturated fatty acids (MUFAs), and three polyunsaturated fatty acids (PUFAs). The *n*-hexane extract showed competitive antioxidant activity with quercetin and the in-silico studies suggested a notable antioxidant activity of the seed oil contents with apparent drug-likeness properties. Further studies are required to separate the extracts, then perform in vitro antioxidant activity on the compounds.

**Keywords:** natural antioxidants; unsaturated fatty acids; radical scavenging compounds; drug-likeness properties; molecular docking study

## 1. Introduction

Reactive oxygen species (ROS) is a commonly used term in biology to describe unstable and highly reactive oxygen-containing chemicals in mammalian cells, including hydrogen peroxide, hydroxyl radicals, superoxide, peroxynitrite, lipid hydroperoxide, singlet oxygen, hypochlorous acid, ozone, alkoxyl radical, and peroxyl radical [1–3]. These molecules are either physiological by-products of the aerobic mitochondrial metabolism or generated as immune responses to pathogens, xenobiotics, and cytokines [4,5]. However, ROS production is necessary for cellular signaling, overproduction of ROS could lead to various tissue and organ failures through damaging lipids, proteins, carbohydrates, and DNA. The diseases linked to ROS include neurodegenerative, digestive, cancer, respiratory, endocrine, cardiovascular diseases, and age-accelerating the processes [5–9]. The sources of ROS production are either exogenous or endogenous. The exogenous triggers to produce ROS could be ionizing radiation, tobacco, alcoholic drinks, pollutants, ultraviolet (UV) radiation, food, and medications. On the other hand, intracellular enzymes are the main sources of endogenous ROS production [10]. Antioxidants are stable molecules that can neutralize free radical molecules in vivo by donating electrons and have the capability to prevent or reduce

cell and tissue damage [11]. The sources of these free radical-scavenging molecules are either endogenous or exogenous. Endogenous antioxidants can be classified into enzymatic and non-enzymatic antioxidants when they are produced by normal cell metabolism. In contrast, exogenous antioxidants are compounds that cannot be produced in the body; therefore, they must be administered either as nutrients or as dietary supplements [12,13].

The human body's defense cells have an efficient capacity to neutralize overproduced ROS and minimize the harmful effects induced by oxidative stress [14,15]. Nevertheless, the efficiency of this system could be compromised by several factors, such as diet, age, lifestyle, and diseases. Therefore, antioxidants could be essential to keep the ROS/defense system balance in equilibrium [9,16]. Currently, scientific communities have focused on the nutrient antioxidants due to the side effects connected to the consumption of dietary supplements [17]. The plant sources of antioxidant compounds are herbs, fruits, food products, seeds, spices, and vegetables [18]. Plants contain various hydrophilic and lipophilic phytochemicals to use as antioxidants [19]. Hydrophilic antioxidant molecules include anthocyanins, phenolics, and ascorbic acid, while lipophilic bioactive antioxidant compounds include chlorophylls, tocopherols, carotenoids, and unsaturated fatty acids (UFAs), which are mono and polyunsaturated fatty acids (MUFAs and PUFAs) [20–22]. Plant oils contain three types of fatty acids: saturated fatty acids (SFAs), MUFAs, and PUFAs. MUFAs are fatty acids with a chain containing one C-C double bond due to a two-hydrogen atom deficit [23]. They have health benefits because they could reduce insulin resistance, inflammation, oxidative stress, and dermatoheliosis [24]. Likewise, PUFAs are known as good and essential fats because they have significant health benefits through the regulation of cellular activities and antioxidant properties [25,26]. PUFAs are amphipathic organic compounds with long hydrocarbon chains, contain two or more carbon–carbon double bond and end with carboxylic acid (Figure 1). They are mainly omega-3 and omega-6 fatty acids, which cannot be produced endogenously. The difference between both omega 3 and 6 is referred to the last carbon-carbon double bond location [27]. The antioxidant activity of MUFAs and PUFAs could be through the inhibition of inflammatory responses, prevention of platelet aggregation, and the protection of vascular endothelial cells, cardiac cells, and neurons from damage [28–31].

Phthalate esters, or phthalic acid esters, are lipophilic chemicals either chemically synthesized or obtained in plants and microorganisms (Figure 1). Synthesized phthalate ester derivatives are commonly used in the plastic industry to improve the flexibility and plasticity of synthetic resins, but they are hazardous to the environment. Conversely, biosynthesized chemicals are biologically active compounds that serve as a defense mechanism in living organisms against bacteria, insects, and fungi [32,33]. Numerous phthalate ester derivatives have been isolated from the plants. The chemical structure of the phthalate derivatives reported from natural sources is different from the synthesized molecules, which provide potential biological activities such as anti-inflammatory, cytotoxic, antiviral, antifungal, antibacterial, and antioxidant activities [34]. According to the studies conducted by Qian et al., (2012) and Kiros et al., (2022), the isolated phthalate ester derivatives from the plants exhibited noteworthy antioxidant activities [35,36].

**Figure 1.** General structure of saturated fatty acids, MUFAs, PUFAs, and phthalate ester [37,38].

Seed oils are the major plant source of MUFAs and PUFAs [39,40]. One of the UFAs plant sources is the seed of *Prunus mahaleb* L. because it contains conjugated linolenic fatty acids as isomers of octadecatrienoic fatty acids [41,42]. Previous studies have reported diverse chemical compositions of *Prunus mahaleb* L. seed oil according to the cultivated countries, such as Egypt, Greece, Sudan, and Turkey, but to the best of our knowledge, no study has been undertaken on the chemical compositions of this plant in Iraq [41,43–46]. *Prunus mahaleb* L., also known as mahaleb cheery, white mahaleb, English cherry, or wild cherry, is from a species of cherry tree, in the family Rosaceae, and subfamily of Prunoideae. This plant is three-meter heigh with white flowers that bloom in spring, then ripen in mid-summer to produce dark-red, nutritious plums. This fruit has a strong, bitter, and sour taste. It is native to Iran, central and southern Europe, Syria, Turkey, Armenia, Azerbaijan, and Georgia. It was also reported in the north of Iraq (Kurdistan). The tree is heat-friendly and prefers well-drained soils [47–50]. This plant has been used traditionally to treat a variety of conditions. The juicy fruit, with a spherical and small size, has been used as neuroprotective, anti-diabetic, anti-obesity, anti-inflammatory, anticancer, and cardioprotective due to the high contents of anthocyanins. The rounded kernels are the main sources of fatty acids and proteins that have been used traditionally as expectorants, aphrodisiacs, tonics, diuretics, antidiabetics, and antidiarrheals. In addition to the medicinal uses of this plant, it has been used in the production of lotions, liqueurs, fragrances, and as a flavoring agent. Ultimately, the plant's fruit is a potent free radical scavenger [51–54]. The aim of this study was to identify the phytochemical composition of *Prunus mahaleb* seed oil extract (PMSOE) native to the Kurdistan region of Iraq using two different solvents (*n*-hexane and petroleum ether) and compare it with the chemical composition of other countries' extracts. The second goal was to realize the in vitro and in silico antioxidant activities of the total extract and predict the drug-like properties of the identified compounds.

## 2. Materials and Methods

### 2.1. Chemicals and Reagents

The reagent 2,2-diphenyl-1-picrylhydrazil (DPPH), 1 g, was obtained from Sigma Aldrich (Sigma Aldrich, Munich, Germany). Quercetin standard was acquired from Santa Cruz Biotechnology (Santa Cruz Biotechnology, Dallas, TX, USA). Ethanol and methanol, both extra-pure (>99.9%), were purchased from Scharlau (Scharlau, Barcelona, Spain). Tween 80 (polysorbate 80, polyoxyethylene sorbitan monooleate), and *n*-hexane were purchased from Biochem, Cosne-Cours-sur-Loire, France. Petroleum ether 40–60 °C and *n*-heptane were acquired from Chem-Lab, Zedelgem, Belgium. Sodium hydroxide was obtained from Merck, Darmstadt, Germany, sodium chloride from GCC, London, UK, and anhydrous sodium sulphate was obtained from Neutron, Tehran, Iran.

## 2.2. Plant Material and Seed Oil Extraction

The white mahaleb fruits were harvested when fully ripe in mid-summer, in August 2021, in the Zewe-Sulaymaniyah region (north of Iraq). The fruit peel and seeds were separated, then dried in the shade, and kept in airtight plastic bags in the refrigerator until use. The seeds were crushed into coarse particles with an electric grinder (Gosonic, Shenzhen, China) and put in a thimble of Soxhlet apparatus. To assess the fatty acid content as well as the antioxidant capacity, two extracts were prepared separately by using petroleum ether and *n*-hexane, then refluxed for 6 h at 60–80 °C [45,55]. Anhydrous sodium sulphate was used to dry the extract, followed by filtration and solvent removal with a rotary evaporator at 45 °C under vacuum (Rotatory evaporator, Laborota 4000- efficient, Heidolph, Germany). Later, the seed oil was centrifuged (Universal 320, Hettich, Germany) for 10 min at 5000 rpm, and the clear supernatant part was separated and stored in dark vials at −18 °C [43].

## 2.3. Gas Chromatography-Mass Spectrometry (GC-MS) Analysis

The seed's oil fatty acid composition was analyzed using GCMS-QP2010 Ultra by Shimadzu, Kyoto, Japan, which was equipped with a capillary column (30 m by 0.25 mm ID; with 0.25 μm film thickness) (Table 1). The fatty acid methyl ester (FAME) was prepared by heating the sample with methanolic NaOH first and then with BF3 methanol [56]. A total of 5 mL of *n*-heptane was used to recover methyl esters in the organic phase. Finally, a saturated NaCl solution was added to the mixture, and the two layers were separated using a separating funnel. A total of 1 μL of the *n*-heptane phase was injected onto the GC in split mode. Helium gas was used as a carrier with a flow rate of 1.7 mL/min. The oven temperature was initially set at 40 °C for 2 min, then increased to 200 °C at a rate of 30 °C/min, and finally to 280 °C at a rate of 5 °C/min for another 2 min. The mass selective detector was set on scan mode and had a mass range of 50 to 800 *m/z*. The fatty acids were identified using the National Institute of Standards and Technology 20 (NIST 20) mass database.

**Table 1.** Some advantages and disadvantages of the methods used in the study and their comparison with other methods available for the same purpose.

| # | Methods Used in the Study | Other Research Methods | |
|---|---|---|---|
| | | Advantages | Disadvantages |
| 1 | Soxhlet method: To extract seed oil using *n*-hexane and petroleum ether as solvents | - Soxhlet extraction is faster and less solvent-intensive than maceration [57].<br>- The plant content is extracted continuously using a fresh solvent [57].<br>- The solvents have a low boiling point, which in favor of thermolabile compounds [58].<br>- using n-hexane as solvent results in high Yield% of oil compared to mechanical extractions [58]. | - Soxhlet extraction in compare to super critical fluid extraction (SFC), and microwave assisted extraction (MAE) requires a lot of solvents and a prolonged extraction time [57].<br>- Not suitable for thermolabile metabolites [59].<br>- n-hexane poses risks to human health, safety, and the environment, prompting the search for superior alternatives. Due to its similarity to n-hexane and safer handling, ethyl acetate is a possible choice [60], or using mechanical press extraction method [58]. |
| 2 | GC-MS | - Gas chromatography (GC) can separate volatile substances efficiently and quickly [61]. | - The degradation of thermolabile compounds at elevated operating temperatures [57]. |

**Table 1.** *Cont.*

| # | Methods Used in the Study | Other Research Methods | |
|---|---|---|---|
| | | Advantages | Disadvantages |
| 3 | DPPH radical scavenging activity | - The DPPH assay is a reliable and straightforward method for assessing antioxidant scavenging activity. This is due to the stability of the radical compound, which eliminates the need for its generation as required in other radical scavenging assays such as ABTS * [62]. | - The DPPH method is not suitable for assessing plasma antioxidant activity due to protein precipitation in the alcoholic reaction medium [62]. |
| 4 | Molecular docking with AutoDock 4 | - AutoDock 4 is a cheap method that can be used for a large chemical database in a short period of time in compared to MD *-simulation [63]. | - In AutoDock 4 the receptor is rigid, and the ligand is flexible, but it is necessary to consider the flexibility of both the ligand and receptor, whereby both entities undergo conformational changes to establish a low-energy optimal binding configuration like in MD simulation [63]. |
| 5 | Molecular property and bioactivity prediction in silico | - The primary benefit of conducting in silico studies to predict the pharmacokinetic properties, is the avoidance of useless expenses linked to biological assays of compounds that are likely to exhibit pharmacokinetic issues in the future. This approach can therefore result in significant savings of both time and resources [64]. | - Pharmacokinetic tests conducted both in vitro and in vivo are of major importance in the assessment of novel drugs and are deemed essential [64]. |

* MD is abbreviation for molecular dynamics, and ABTS is abbreviation for 2,2′-azino-bis(3-ethylbenzothiazoline-6-sulfonic acid).

### 2.4. Measurement of Radical Scavenging Activity

The antioxidant capability of both samples was assayed by DPPH according to the procedure explained by Blois with minor adjustments [65,66] (Table 1). The DPPH solution was prepared by dissolving 24 mg of DPPH in 100 mL of 95% ethanol. The seed oil was prepared in a dilution series from 10% mL/mL to 60% mL/mL in 95% ethanol with tween 80 used as a surfactant [66]. Quercetin was used as a positive control with a dilution series of 10–60 μg/mL; 50 μL of each sample and the positive control were transferred into a 96-well plate, and then 150 μL of DPPH solution was added. Incubated in the dark within a shaking incubator (EN61009, Lab-Tech, Namyangju, Republic of Korea) for 30 min at 25 °C. Finally, the absorbance was read at 490 nm by a microplate reader (ELx800, Bio Tek, Winooski, VT, USA), and the radical scavenging activity (%) was calculated according to the equation:

$$[(A\ blank - A\ sample)/A\ blank] \times 100$$

where A blank was the DPPH solution absorbance without sample and A sample was the DPPH solution absorbance with sample [67]. All the procedures were performed in triplicate.

### 2.5. Molecular Docking

In this study, the two- and three-dimensional structures of the identified molecules by GC-MS were drawn using MarvinSketch. Then, the energy minimization algorithms were performed with a MMFF94 forcefield (Marvin version 22.19, ChemAxon; https://chemaxon.com/products/marvin, accessed on 6 February 2023). All the compounds were saved as a PDB file to be ready for molecular docking [68,69]. All protein targets were downloaded from the RCSB Protein Data Bank website in PDB format [70] (https://www.rcsb.org, accessed on 6 February 2023; access codes 2CDU, 3O8Y, 6RKB, 7LAE, and 7TSH) [71–75]. UCSF Chimera version 1.15 was used to remove the ligand, water molecules, and all chains, except chain A; however, chains A and B were kept in myeloperoxidase. Afterward, the clean protein structures were saved as a PDB file to be prepared for molecular docking [76]. The AMDock program (AutoDock4 version 1.5.2) was applied to predict the binding free energy, initiated with the addition of Kollman charges and polar hydrogen. Then, the files were saved as a pdbqt file. AMDock uses PyMOL to view molecular structures, starting with many predefined visualization schemes to set up the appropriate box, define the search space, visualize, and realize the docking results [77,78]. As shown in Table 2, prior to docking, the grid box for the search space of the selected targets was determined with the AutoDock tool (version 1.5.7). The binding affinities between the ligands and receptors were predicted and ranked using AutoDock4.

**Table 2.** The parameters of each protein target in the molecular docking studies.

| Protein | PDB ID | Resolution Å | Grid Box Center | Grid Box Size | Ref. |
|---|---|---|---|---|---|
| NADPH oxidase | 2cdu | 1.8 | x center = 6.9<br>y center = −1.5<br>z center = 2.8 | x-dimension = 54<br>y-dimension = 58<br>z-dimension = 46 | [14] |
| 5-Lipoxygenase | 3o8y | 2.39 | x center = −4.3<br>y center = 15.7<br>z center = 5.9 | x-dimension = 72<br>y-dimension = 46<br>z-dimension = 52 | [15] |
| Monoamine oxidase B | 6rkb | 2.3 | x center = 55<br>y center = 155.2<br>z center = 30.3 | x-dimension = 56<br>y-dimension = 68<br>z-dimension = 56 | [16] |
| Myeloperoxidase | 7lae | 2.97 | x center = −9.3<br>y center = 21.5<br>z center = −20.6 | x-dimension = 84<br>y-dimension = 66<br>z-dimension = 54 | [17] |
| Nitric oxide synthase | 7tsh | 2.15 | x center = 51.6<br>y center = 31.6<br>z center = −188.3 | x-dimension = 70<br>y-dimension = 84<br>z-dimension = 76 | [18] |

### 2.6. Molecular Property and Bioactivity Prediction

The physicochemical and drug-liken properties of the molecules are directly linked to their bioactivity [79]. The drug-likeness properties and the bioactivity scores of the twelve selected compounds and the standard molecule (quercetin) were investigated using the free online software Molinspiration Cheminformatics (Molinspiration v2022.08., https://www.molinspiration.com/cgi-bin/properties, accessed on 5 March 2023) [80,81].

### 2.7. Statistical Analysis

The radical scavenging activity values of the seed oil samples against DPPH were calculated as a mean ± standard deviation of triplicated measurements for each test. This study was to realize nonlinear correlation analysis between different concentrations of each sample and the percentage of DPPH inhibition using GraphPad Prism Version 9.5.1 (Graph-Pad Software Inc., San Diego, CA, USA; www.graphpad.com, accessed on 5 March 2023). In this analysis, the Pearson correlation coefficient (R) and coefficient of determination ($R^2$) were calculated because we assumed that both concentration and the percentage of

inhibition values followed a Gaussian distribution. Two-tailed *p* values were calculated, as they are more accurate and could correctly estimate the direction of the difference for most of the statistical data. Furthermore, the confidence interval was 95% [82]. Additionally, the predicted values of the percentage of inhibition depending on the concentration and the strength of the predictions were calculated by simple linear regression analysis which is a fundamental and widely utilized method of predictive analysis for estimating scores on one variable based on the scores of a second variable [82,83].

## 3. Results and Discussion

### 3.1. Extraction of The Seed Oil of Prunus mahaleb L. Using Different Solvents

Oil can be extracted from plant seeds using various solvents, such as ethanol, isopropanol, chloroform, *n*-hexane, diethyl ether, petroleum ether, and *n*-heptane [84,85]. Selecting a solvent is the key step during extraction because different solvents presumably provide different compositions and yields [47,86]. In this research, two different solvents were used to extract oil from the seed of *Prunus mahaleb* L., which were petroleum ether and *n*-hexane, since *n*-hexane is one of the most common solvents to extract oils from plant sources due to the miscibility of the oils in this solvent and easy recovery [58]. However, both solvents have a low polarity index (0.1) to extract the seed oil and a low boiling temperature, which is in favor of the heat-sensitive compounds [87]. In this study, the percentage (g/100 g) of *Prunus mahaleb* L. seed oil extracted with petroleum ether (PMSOEPE) was (30.5%), which was approximately equal to the yield provided by Mariod et al., (2009) [45]. However, the contents of the extracted oil were significantly higher than the previous works achieved by Özgül-Yücel, (2005) (4.7–18.5%) [46] and Johansson et al. (1997) (12–16%) [88]. The yield of *Prunus mahaleb* L. seed oil extracted with *n*-hexane (PMSOENH) was (26%), while the earlier studies by Sbihi, (2014) and Sbihi et al., (2015) reported a slightly higher percentage of the extract (31%) [41,55]. Finally, PMSOENH was provided with a bright yellowish color compared with the dark yellow PMSOEPE color (Figure 2).

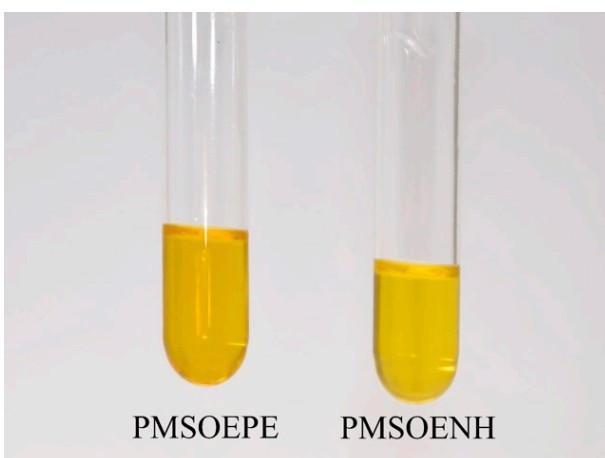

**Figure 2.** Color appearance of PMSOENH and PMSOEPE.

### 3.2. Composition of the Seed Oil

GC-MS is widely used to identify and analyze different compounds, such as hydrophilic, hydrophobic, and volatile compounds [89]. In this study, GC-MS analysis of both PMSOEPE and PMSOENH identified seventeen FAs with different quantities. Oleic acid had the highest percentage of PMSOEPE (37.91%), followed by (32.04%) of 6,9,11-octadecatrienoic acid, and (24.47%) of linoleic acid. The main FAs in PMSOENH were 6,9,11-octadecatrienoic acid (34%), oleic acid (32.11%), and linoleic acid (25.63%), respectively. The remaining FAs had approximately the same yields for both samples, except stearic acid, which was (1.45%) for PMSOEPE and (1.96%) for PMSOENH. Furthermore, palmitoleic acid was unavailable in PMSOEPE, but it was (0.25%) in PMSOENH. The total

PUFAs content of PMOENH was higher (59.82%) than PMSOEPE (56.69%), but the MUFAs content of PMSOENH was lower (32.7%) than PMSOEPE (38.27%). The list of FAs with their retention time (RT) and yield percentage is presented in Table 3.

**Table 3.** Saturated fatty acids, MUFAs, and PUFAs extracted from PMSOENH and PMSOEPE.

| Comp. | IUPAC Name | Common Name | MW (g/mol) | RT (min) | Percentage PMSOE (%) | | % of Similarity by NIST.20 | *m/z* Values |
|---|---|---|---|---|---|---|---|---|
| | | | | | n-hexane | PE ** | | |
| C01 P * | 9,11,13-octadecatrienoic acid | α -eleosteric acid | 278 | 16.09 | 0.19 | 0.18 | 90 | 292, 261, 232, 135, 92, 59 |
| C02 P | 9,12-Octadecadienoic acid | Linoleic acid | 280 | 13.72 | 25.63 | 24.47 | 96 | 294, 263, 137, 97, 59, 57 |
| C03 P | 6,9,11-octadecatrienoic acid | X | 278 | 15.56 | 34.00 | 32.04 | 91 | 292, 177, 137, 71, 66 |
| C04 m * | 9-eicosenoic acid | Gadelaidic acid | 310 | 16.52 | 0.13 | 0.12 | 91 | 324, 293, 250, 208, 127, 85 |
| C05 m | 9-heptadecenoic acid | Margaroleic acid | 268 | 12.54 | 0.04 | 0.03 | 90 | 282, 251, 125, 85, 59 |
| C06 m | 9-Octadecenoic acid | Oleic acid | 282 | 13.85 | 32.11 | 37.91 | 96 | 296, 265, 99, 59, 41 |
| C07 m | 11-hexadecenoic acid | Palmitvaccenic acid | 254 | 11.45 | 0.17 | 0.19 | 94 | 268, 237, 83, 59, 41 |
| C08 m | 9-Hexadecenoic acid | Palmitiolic acid | 254 | 12.00 | 0.25 | X | 91 | 268, 237, 111, 71, 59, 41 |
| C13 s * | Hexadecanoic acid | Palmitic acid | 256 | 11.70 | 3.10 | 3.00 | 97 | 270, 239, 57, 43, 29 |
| C14 s | Octadecanoic acid | Stearic acid | 284 | 14.14 | 1.96 | 1.45 | 97 | 298, 267, 224, 59, 57, 43 |
| C15 s | Eicosanoic acid | Arachidic acid | 312 | 16.93 | 0.28 | 0.27 | 95 | 326, 295, 59, 57, 43, 29 |
| C16 s | Docosanoic acid | Behenic acid | 340 | 19.85 | 0.08 | 0.06 | 93 | 354, 323, 71, 59,57, 43, 29 |
| C17 s | Heptadecanoic acid | Margaric acid | 270 | 12.85 | 0.04 | 0.04 | 91 | 284, 253, 59, 57, 43, 29 |
| C18 s | Tetracosanoic acid | Lignoceric acid | 368 | 22.74 | 0.06 | 0.05 | 91 | 382, 351, 71, 59, 57, 43 |
| PUFA | | | | | 59.82 | 56.69 | | |
| MUFA | | | | | 32.70 | 38.25 | | |
| SFA | | | | | 5.62 | 4.87 | | |

* m, p, and s superscripts mean MUFAs, PUFAs, and saturated fatty acids. ** PE abbreviation for petroleum ether.

According to the literature, the FA constituents and yield percentages of PMSOE are significantly different. For example, Alma et al. (2011) used the cold press method to extract the major constituents of Turkish PMSOE. The main FAs obtained were (35.8%) oleic acid, (24.9%) linoleic acid, and (22.6%) linolelaidic acid [43]. Another example is the study performed by Mariod et al., (2009) on Sudanese PMSOE using petroleum ether as a solvent at 40–60 °C and Soxhlet for extraction. The reported essential yields of FAs were (45%) oleic acid, (47%) linoleic acid, (5.7%) palmitic acid, and (1.3%) stearic acid [45]. However, the main reported FAs contents of Syrian PMSO using *n*-hexane solvent and Soxhlet apparatus were (40.7%) alpha-eleostearic acid, (29.8%) oleic acid, and (26.6%) linoleic acid [41]. Furthermore, Mead et al., (2016) extracted Egyptian PMSOE using the maceration technique and petroleum ether as a solvent at 60–80 °C. The results showed timnodonic acid as a major component (33.1%), followed by oleic acid (28.7%), and linoleic acid (24.4%) [90]. The results of our study showed three FAs (palmitvaccenic acid, gadelaidic acid, and 6,9,11-octadecatrienoic acid) that were not obtained in earlier studies. Moreover, the highest constituent was 6,9,11-octadecatrienoic acid. In addition, PMSOENH contained a higher quantity of PUFA (59.82%) relative to the Sudanese, Turkish, and Egyptian PMSOE (47.1%, 56.1%, and 57.79%, respectively) (Table 4).

**Table 4.** Comparison among the various studies content extracts of FAs.

| Common Name | Percentage of Fatty Acid Composition % | | | | |
|---|---|---|---|---|---|
| | Egypt (Maceration) | Iraq (Soxhlet) | Sudan, (Soxhlet) | Syria (Soxhlet) | Turkey (Cold Press) |
| Lauric acid | - | - | < 0.1 [pe] * | - | - |
| Myristic acid | - | - | < 0.1 [pe] | 0.04 [n] ± 0.01 | 0.1 [c] * |
| Palmitic acid | 2.74 [pe] | 3.1 [n] *, 3.0 [pe] | 5.7 [pe] ± 0.02 | 3.84 [n] ± 0.05 | 5.6 [c] |
| Palmitiolic acid | 0.17 [pe] | 0.25 [n] | - | 0.23 [n] ± 0.01 | 0.5 [c] |
| Palmitvaccenic acid | - | 0.17 [n], 0.19 [pe] | - | - | - |
| Margaric acid | 0.06 [pe] | 0.04 [n], 0.04 [pe] | - | - | - |
| Margaroleic acid | 0.06 [pe] | 0.04 [n], 0.03 [pe] | - | - | - |
| Stearic acid | 1.73 [pe] | 1.96 [n], 1.45 [pe] | 1.3 [pe] ± 0.3 | 1.88 [n] ± 0.04 | 2.2 [c] |
| Oleic acid | 28.71 [pe] | 32.11 [n], 37.91 [pe] | 45 [pe] ± 0.5 | 29.83 [n] ± 0.5 | 35.8 [c] |
| Cis-vaccenic acid | - | - | - | 0.67 [n] ± 0.04 | - |
| Linoleic acid | 24.35 [pe] | 25.63 [n], 24.47 [pe] | 47 [pe] ± 0.5 | 21.68 [n] ± 0.4 | 24.9 [c] |
| Linolelaidic acid | - | - | - | - | 22.6 [c] |
| $\alpha$-Linoleic acid | - | - | - | - | 3 [c] |
| 6,9,11-octadecatrienoic acid | - | 34.00 [n], 32.04 [pe] | - | - | - |
| $\alpha$-eleosteric acid | - | 0.19 [n], 0.18 [pe] | - | 40.71 [n] ± 0.8 | - |
| $\alpha$-Linolenic acid | 0.37 [pe] | - | 0.1 [pe] ± 0.02 | - | - |
| Arachidic acid | 0.73 [pe] | 0.28 [n], 0.27 [pe] | - | 0.33 [n] ± 0.01 | 0.5 [c] |
| Gadelaidic acid | - | 0.13 [n], 0.12 [pe] | - | - | - |
| Gadoleic acid | 0.41 [pe] | - | - | 0.43 [n] ± 0.03 | 0.3 [c] |
| Eicosadienoic acid | - | - | - | 0.29 [n] ± 0.01 | 0.3 [c] |
| Timnodonic acid | 33.07 [pe] | - | - | - | - |
| Behenic acid | 0.72 [pe] | 0.08 [n], 0.06 [pe] | - | 0.4 [n] ± 0.01 | 0.3 [c] |
| Erucic acid | 6.74 [pe] | - | - | - | - |
| Lignoceric acid | 0.14 [pe] | 0.06 [n], 0.05 [pe] | - | - | 0.7 [c] |
| SFA | 6.12 | 5.62 [n], 4.87 [pe] | 7.2 | 6.49 ± 0.12 | 9.4 |
| MUFA | 36.09 | 32.7 [n], 38.25 [pe] | 45 ± 0.5 | 31.16 ± 0.58 | 36.6 |
| PUFA | 57.79 | 59.82 [n], 56.69 [pe] | 47.1 ± 0.5 | 62.68 ± 1.21 | 52.1 |

* c superscript means extraction performed with cold press; n superscript means extraction performed with *n*-Hexane solvent; pe superscript means extraction performed with petroleum ether solvent.

In addition to FAs, four phthalate ester derivatives were obtained from PMSOENH: 1,2-diethyl benzene-1,2-dicarboxylate (0.26%), 1-(2-ethylhexyl) 2-methyl benzene-1,2-dicarboxylate (0.07%), 1,2-bis(2-ethylhexyl) benzene-1,2-dicarboxylate (0.43%), and 1,2-bis-2methylpropyl benzene-1,2-dicarboxylate (0.33%), but they were not identified in the PMSOEPT analysis (Table 5), which might be due to the differences in solvent polarity.

**Table 5.** Phthalate derivatives found in PMSOENH.

| Comp. | IUPAC Name | Common Name | MW (g/mole) | Retention Time (min) | Percentage (%) | % of Similarity by NIST.20 | *m/z* Values |
|---|---|---|---|---|---|---|---|
| **C09** | 1,2-diethyl benzene-1,2-dicarboxylate | Diethyl Phthalate | 222 | 8.69 | 0.26 | 89 | 222, 177, 150, 132, 76 |
| **C10** | 1-(2-ethylhexyl) 2-methyl benzene-1,2-dicarboxylate | 1-2-ethylhexyl2-methylphthalate | 292 | 12.68 | 0.07 | 87 | 292, 150, 92, 76 |
| **C11** | 1,2-bis(2-ethylhexyl) benzene-1,2-dicarboxylate | Etalon | 390 | 19.76 | 0.43 | 97 | 390, 261, 132, 76, 29 |
| **C12** | 1,2-bis-2methylpropyl benzene-1,2-dicarboxylate | Diisobutyl pthalate | 278 | 10.91 | 0.33 | 94 | 278, 205, 132, 76, 29 |

### 3.3. In Vitro Antioxidant Activity (DPPH Free Radical Scavenging Activity)

This assay was applied to determine the total antioxidant activity using antiradical tested molecules with the stable free radical DPPH to produce 1,1-diphenyl-2-picrylhydrazine. The principle of this assay is based on the discoloration due to a reduction in the antiradical substance [89]. The level of discoloration indicates the compounds' capacity to donate hydrogen or electrons to neutralize the free radicals [40]. This technique can be used to realize the antioxidant capacity of various compounds, including plant extracts,

fruits, and juices [91–93]. According to the literature, increasing antioxidant concentration increases antioxidant capacity [94–96]. This experiment confirmed the antioxidant activity of both PMSOENH and PMSOEPE, also confirmed that the radical scavenging activity had a linear correlation with the extract's concentration, when the antioxidant activity was calculated as a percentage of DPPH scavenging activity. The antioxidant capacities of quercetin, PMSOENH, and PMSOEPE were strongly correlated with their concentrations because the coefficient of determination ($R^2$) of quercetin was (0.9), PMSOENH was (0.92), and PMSOEPE was (0.95). Furthermore, according to linear regression analysis, quercetin ($y = 1.302x + 8.566$, $R^2 = 0.90$), PMSOENH ($y = 1.269x + 3.490$, $R^2 = 0.91$), and PMSOEPE ($y = 0.8948x - 2.904$, $R^2 = 0.93$) had concentrations that were significantly correlated with the estimated percent of inhibition. As shown in Figure 3, the maximum antioxidant activities of PMSOENH and PMSOEPE at a concentration of 60 mL/mL were (68.37%) and (45.48%), respectively. In addition, the maximum antioxidant activity of the positive control at a concentration of 60 mcg/mL was (72.70%).

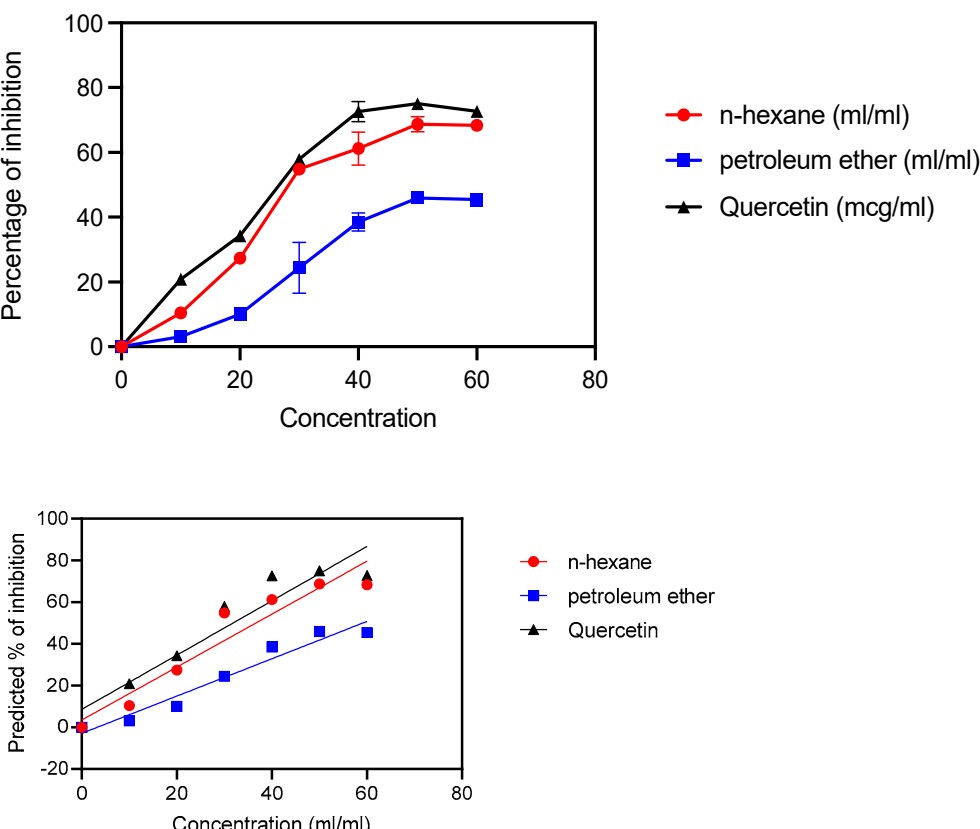

**Figure 3.** Antioxidant activity of quercetin, PMSOENH, and PMSOEPE using DPPH free radical scavenging rate.

### 3.4. In Silico Investigations (Molecular Docking)

Free radicals are produced endogenously through several enzymes, such as NADPH oxidase (NOX) [97], nitric oxide synthase (NOS) [98], myeloperoxidase (MPO) [99], 5-lipoxygenase (5-LOX) [100], monoamine oxidase B (MAOB) [101], xanthine oxidase [102], and cyclooxygenase [103]. However, free radical production by various enzymes is necessary for the physiological and biological functions; overproduction of such substances could produce harmful oxidative stress, which should be neutralized through antioxidant defense mechanisms or nutraceutical antioxidants [104]. In silico studies were performed to realize the antioxidant effects of the extracted oil from white mahaleb seed through the prediction of binding interactions and inhibitions of the enzymes that have a critical role in intracellular oxidations, such as NOX, NOS, MPO, 5-LOX, and MAOB [105,106]. The seed

oil composition of PMSOENH and PMSOEPE was mainly FAs, but four phthalate derivatives were also extracted from PMSOENH (Tables 3 and 5). In addition to the verifiable antioxidant activity of FAs [107], natural phthalate derivatives were recorded as inhibitors of the oxidative enzymes [35].

Binding free energies between the compounds and the selected enzymes were represented by $\Delta G$ values (kcal/mol). In this study, the 12 extracted compounds (eight UFAs and four phthalate derivatives) and the positive control were docked against the five selected enzymes; therefore, the total calculated $\Delta G$ values were 70. As presented in Table 6. The positive control had the highest binding free energies to 5-LOX ($-7.81$ kcal/mol) and MAOB ($-8.65$ kcal/mol), and the second highest binding free energies to NOX ($-6.87$ kcal/mol) and MPO ($-8.38$ kcal/mol), after **C10**. However, it had the fifth rank to NOS ($-6.59$ kcal/mol). The most promising results of docking were for **C10**, which had the highest $\Delta G$ to NOX ($-7.27$ kcal/mol), MPO ($-8.6$ kcal/mol), and NOS ($-7.98$ kcal/mol), and the second highest binding affinity to 5-LOX and MAOB ($-7.5$ and $-8.52$ kcal, respectively). According to exploring the binding free energy of each compound, the highest $\Delta G$ of **C01**, **C03**, **C05**, **C06**, **C07**, **C08**, and quercetin were to MAOB; the highest $\Delta G$ of **C09**, **C10**, **C02**, and **C04** were to MPO. Finally, the highest $\Delta G$ of the remaining compounds (**C11** and **C12**) were to NOS ($-7.21$ and $-7.5$ kcal/mol, respectively).

**Table 6.** The chemical formula and binding free energy of the selected ligands.

| Compound | Chemical Formula | $\Delta G$ (kcal/mol) | | | | |
| --- | --- | --- | --- | --- | --- | --- |
| | | NOX | 5$-$LOX | MAOB | MPO | NOS |
| **C01** | $C_{18}H_{30}O_2$ | $-5.78$ | $-4.79$ | $-6.90$ | $-5.34$ | $-4.99$ |
| **C02** | $C_{18}H_{32}O_2$ | $-4.99$ | $-3.81$ | $-6.71$ | $-7.17$ | $-4.24$ |
| **C03** | $C_{18}H_{30}O_2$ | $-5.74$ | $-5.81$ | $-7.18$ | $-6.60$ | $-6.89$ |
| **C04** | $C_{20}H_{38}O_2$ | $-6.54$ | $-4.58$ | $-4.39$ | $-7.04$ | $-5.85$ |
| **C05** | $C_{17}H_{32}O_2$ | $-4.79$ | $-5.96$ | $-7.55$ | $-5.53$ | $-5.78$ |
| **C06** | $C_{18}H_{34}O_2$ | $-4.75$ | $-5.87$ | $-7.05$ | $-6.17$ | $-5.30$ |
| **C07** | $C_{16}H_{30}O_2$ | $-6.16$ | $-4.84$ | $-6.84$ | $-5.31$ | $-5.40$ |
| **C08** | $C_{16}H_{30}O_2$ | $-5.35$ | $-5.10$ | $-7.13$ | $-6.06$ | $-5.77$ |
| **C09** | $C_{12}H_{14}O_4$ | $-5.52$ | $-6.30$ | $-6.03$ | $-6.40$ | $-5.77$ |
| **C10** | $C_{17}H_{24}O_4$ | $-7.27$ | $-7.50$ | $-8.52$ | $-8.60$ | $-7.98$ |
| **C11** | $C_{24}H_{38}O_4$ | $-6.35$ | $-4.74$ | $-5.09$ | $-4.56$ | $-7.21$ |
| **C12** | $C_{16}H_{22}O_4$ | $-5.87$ | $-6.91$ | $-6.48$ | $-6.05$ | $-7.50$ |
| **Quercetin** | $C_{15}H_{10}O_7$ | $-6.87$ | $-7.81$ | $-8.65$ | $-8.38$ | $-6.59$ |

Small molecules (MW < 500 Da) are the preferred form of biomolecules and therapeutics because of their favorable pharmacokinetic features and oral bioavailability [108]. The oral bioavailability of the compounds can be determined through Lipinski's rule of five criteria [109,110]. This rule is based on the physicochemical properties of the compounds and states that the compounds to be orally absorbed and permeated through intestinal membranes should have the following criteria: MW $\leq$ 500 Da, logP $\leq$ 5, H-bond donors and acceptors $\leq$5 and 10, respectively. Any molecule that violates more than one of the mentioned criteria could appear to have absorption or permeation issues and eventually may not be orally bioavailable [111]. Moreover, rotatable bonds (RB) or molecular flexibility, is one of the parameters used to predict drug-like properties because exceeding 10 RB may limit oral bioavailability [112]. Lastly, topological polar surface area (TPSA) of the molecule is another indicator of the compound's absorption because the TPSA of orally bioavailable drugs should not exceed 120 Å [113].

In the current study, the physicochemical properties, or drug-likeness properties, of all the PMSOE compounds were calculated through Molinspiration software (Molinspiration v2022.08) to explore the correlations between antioxidant activity and oral bioavailability of the compounds [114]. All the compounds passed Lipinski's rule of five, because none of the compounds exhibited more than one criteria violation. Compounds **C09**, **C10** and **C12** had no violations, but the remaining compounds had only one violation. Most of

the compounds were highly lipophilic (logP > 5) because they were FAs, except phthalate derivatives (**C09**, **C10**, and **C12**), and quercetin. The other molecular descriptors of all compounds, such as MW, HB donor, and HB acceptor, were in the normal range. However, all the compounds were highly flexible (violating the critical limit of RB), except the compounds **C09**, **C10**, **C12**, and quercetin. Finally, all the extracted compounds obeyed the TPSA normal range, all but the standard compound disobeyed the limit (Table 7).

**Table 7.** Lipinski's rule of five criteria calculations of the PMSOE and the standard compound utilizing molinspiration software.

| Compound | milogP | MW | HB Acceptor | HB Donor | Violations | RB | TPSA |
|---|---|---|---|---|---|---|---|
| **C01** | 6.60 | 278.44 | 2 | 1 | 1 | 13 | 37.30 |
| **C02** | 6.86 | 280.45 | 2 | 1 | 1 | 14 | 37.30 |
| **C03** | 6.37 | 278.44 | 2 | 1 | 1 | 13 | 37.30 |
| **C04** | 8.47 | 310.52 | 2 | 1 | 1 | 17 | 37.30 |
| **C05** | 7.08 | 268.44 | 2 | 1 | 1 | 14 | 37.30 |
| **C06** | 7.58 | 282.47 | 2 | 1 | 1 | 15 | 37.30 |
| **C07** | 6.57 | 254.41 | 2 | 1 | 1 | 13 | 37.30 |
| **C08** | 6.57 | 254.41 | 2 | 1 | 1 | 13 | 37.30 |
| **C09** | 2.31 | 222.24 | 4 | 0 | 0 | 6 | 52.61 |
| **C10** | 4.75 | 292.38 | 4 | 0 | 0 | 10 | 52.61 |
| **C11** | 7.94 | 390.56 | 4 | 0 | 1 | 16 | 52.61 |
| **C12** | 3.80 | 278.35 | 4 | 0 | 0 | 8 | 52.61 |
| **Quercetin** | 1.68 | 302.24 | 7 | 5 | 0 | 1 | 131.35 |

As shown in Table 8, the bioactivity scores were calculated for the phthalate derivatives, FAs, and the standard molecules targeting enzymes, proteases, nuclear receptor, kinases, ion channels, and GPCR. The activities of the compounds are predicted through the calculated bioactivity scores. For instance, scores larger than 0 are probably biologically active, while values −0.5–0 mean that the compound is moderately active. On the other hand, bioactivity scores below −0.5 are expected to be inactive [115,116]. In earlier studies, the correlation between ion channels and diseases related to oxidative stress was predominantly documented in the context of cardiovascular and neurodegenerative pathologies. Several research studies have shown the participation of potassium, sodium, calcium, and chloride channels in the pathogenesis of diseases characterized by significant oxidative stress [117]. The studied FAs **C01**–**C08** showed activity on ion channels modulator, which suggests promising antioxidant capability. According to the results predicted by Molinspiration, all compounds were moderately active on kinase inhibitors, except **C09** and quercetin, which were inactive and active, respectively. In addition, all the compounds and quercetin were active enzyme inhibitors, except **C09**–**C012**, which were moderately active. Compound **C09** was inactive; **C10**–**C12**, **C05**, **C07**, **C08**, and quercetin were moderately active; **C01**–**C04**, and **C06** were active on protease inhibitor. Furthermore, all the compounds and the standard molecule were active on nuclear receptor ligand, but only **C09** was inactive, and both **C10** and **C12** were moderately active. The activities of the compounds on the GPCR ligand were approximately the same; the difference was in the activity of **C09**, which was inactive. In conclusion, the phthalate derivatives (**C09**–**C12**) were moderately inhibiting enzymes, while all the FAs and quercetin were presumably inhibiting the enzymes (Table 8).

**Table 8.** Bioactivity calculations of the PMSOE compounds and quercetin using Molinspiration software.

| Compound | GPCR Ligand | Ion Channel Modulator | Kinase Inhibitor | Nuclear Receptor Ligand | Protease Inhibitor | Enzyme Inhibitor |
|---|---|---|---|---|---|---|
| C01 | 0.2 | 0.1 | −0.2 | 0.3 | 0.1 | 0.3 |
| C02 | 0.3 | 0.2 | −0.2 | 0.3 | 0.1 | 0.4 |
| C03 | 0.3 | 0.2 | −0.1 | 0.4 | 0.1 | 0.4 |
| C04 | 0.2 | 0.1 | −0.1 | 0.3 | 0.1 | 0.3 |
| C05 | 0.1 | 0.1 | −0.3 | 0.2 | 0.0 | 0.3 |
| C06 | 0.2 | 0.1 | −0.2 | 0.2 | 0.1 | 0.3 |
| C07 | 0.1 | 0.1 | −0.4 | 0.1 | 0.0 | 0.3 |
| C08 | 0.1 | 0.1 | −0.4 | 0.1 | 0.0 | 0.3 |
| C09 | −0.6 | −0.2 | −0.7 | −0.5 | −0.7 | −0.3 |
| C10 | −0.1 | −0.1 | −0.3 | 0.0 | −0.1 | −0.1 |
| C11 | 0.0 | 0.0 | −0.1 | 0.1 | 0.0 | 0.0 |
| C12 | −0.2 | −0.1 | −0.3 | −0.1 | −0.2 | −0.1 |
| Quercetin | −0.1 | −0.2 | 0.3 | 0.4 | −0.3 | 0.3 |

## 4. Conclusions

The results obtained from the study suggest that *Prunus mahaleb* L. possesses nutritional and phytochemical metabolites that exhibit significant health-promoting properties. The PMSOE from the Kurdistan region of Iraq explored MUFAs, PUFAs, and phthalate derivatives. Some of the ingredients differed from PMSOEs from other countries reported in previous studies. The diversity of FA types and contents could be due to different geographical locations, extraction techniques, genetics, climate, and solvents. Both PMSOEPE and PMSOENH had antioxidant activity, but the latter exhibited higher antioxidant activity, competitive with quercetin, which might be either due to the higher concentration of PUFAs or the availability of phthalate ester derivatives. Molecular docking calculations further predicted the antioxidant potential of all compounds against the five free radical-producing intracellular enzymes; **C10** had the highest binding score against all the tested enzymes; however, additional studies like molecular dynamics simulation should be applied to obtain a more precise estimation of ranking the binding score against all the tested enzymes [118]. Lastly, the calculated drug-likeness property and bioactivity exhibited that all compounds were in accordance with Lipinski's rule of five; the bioactivity score against the target molecules exhibited that **C09–C12** were moderately active and **C01–C08** were active on the enzyme inhibitors. In addition, the bioactivity scores obtained in ion channel modulators for the identified FAs suggest their probability as good antioxidant supplements to be studied in diseases linked to ROS. Further studies to separate the seed oil contents of *Prunus mahaleb* L. and perform radical scavenging with enzyme inhibitory activity to investigate the actual antioxidant capacity of each compound are highly warranted.

**Author Contributions:** Conceptualization, Z.N.H.; methodology, Z.N.H.; validation, H.A.A. and T.S.; formal analysis, Z.N.H., H.A.A. and T.S.; investigation, Z.N.H.; resources, Z.N.H.; data curation, Z.N.H. and T.S.; writing—original draft, Z.N.H.; writing-review, and editing; Z.N.H., H.A.A. and T.S.; visualization, Z.N.H., H.A.A. and T.S.; supervision, H.A.A. and T.S.; project administration, Z.N.H., funding acquisition, Z.N.H., H.A.A. and T.S. All authors have read and agreed to the published version of the manuscript.

**Funding:** This research received no external funding.

**Institutional Review Board Statement:** Not applicable.

**Informed Consent Statement:** Not applicable.

**Data Availability Statement:** The data are contained within the article available at https://zenodo.org/record/7991880, accessed on 31 May 2023.

**Acknowledgments:** The authors appreciate the support from the College of Pharmacy at the University of Sulaimani and are thankful for the guidance of Saman Abdulrahman Ahmed/College of Agriculture, University of Sulaimani in Sulaymaniyah, Iraq.

**Conflicts of Interest:** The authors declare no conflict of interest.

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
