# Peer review of "Antioxidant Activity of the Prunus mahaleb Seed Oil Extracts Using n-Hexane and Petroleum Ether Solvents: In Silico and In Vitro Studies"

_applsci, doi:10.3390/app13137430_

Round 1
Reviewer 1 Report
Antioxidant Activity of the Prunus mahaleb Seed Oil Extracts Using n-Hexane and Petroleum Ether Solvents: In Silico and In Vitro Studies
Z. Hussein, H.A. Azeez, & Twana Salih
The manuscript entitled “Antioxidant Activity of the Prunus mahaleb Seed Oil Extracts Using n-Hexane and Petroleum Ether Solvents: In Silico and In Vitro Studies” by Hussein et al. shows the antioxidant activity and potential clinical use of fatty acids and phthalate esters from Prunus mahaleb.
Although the work is fascinating, some aspects must be corrected before publication. For example:
a) Natural products is no longer a new research field. All (hard) data should be curated in databanks (Figshare, Metabolights, etc.) for researchers worldwide to look better at the results.
b) In Table 2 and Table 4, the m/z values used for quantification and identification should also be presented (in addition to the chemical formula and the RT). Furthermore, the similarity % given by the NIST database to identify each compound.
c) I would also suggest the authors start the Comp. codes C01 in Table 2. For example, The first compound in Table 2 is 9,11,13-octadecatrienoic acid (alpha-eleosteric acid). Nevertheless, it is denominated as compound 5. While, Diethyl Phthalate, the first entrance of Table 4, is denominated compound 1.
In addition:
Line 387-388. In conclusion, the phthalate derivatives (C01–C04) were moderately inhibiting enzymes, while all the FAs and quercetin were presumably inhibiting the enzymes (Table 7).
Although I agree with the authors that physicochemical or drug-likeness properties can be derived theoretically, I would state that ADME measurements (in-vitro / in-vivo) are always required to demonstrate the compounds' water (bioavailability), diffusion/transport, clearance, etc. Thus, stating that the compounds identified by the authors would have moderate activity on an ion channel modulation without any in-detailed biological assay is highly controversial.
Author Response
We would like to thank the reviewers for taking their precious time reviewing our manuscript. We highly appreciate their suggestions and comments. Please see the attachment below for viewing our responses to the reviewer.

Reviewer 2 Report
In this work, the authors studied 12 PMSOE compounds for their druggability properties. My biggest concern is the interpretation of docking scores. A docking score is not equivalent to a binding free energy. While docking scores can be used to rank docked poses but a better docking score is not necessarily indicates a better binding affinity. In fact, it has been proved by many previous studies that using docking scores to rank the compounds for their binding potency is not reliable. (see https://doi.org/10.1002/cmdc.202200425, https://doi.org/10.1021/jm050362n, https://doi.org/10.3390/molecules23081899) The authors should clarify this in the manuscript when talking about docking scores. If the authors can provide experimental data to support their predictions from docking scores, I strongly suggest the authors to provide such data. Otherwise, the authors should highlight the limitation of using docking scores to predicte the binding affinity of studied compounds with references of these previous studies (or even more related work) to make sure readers do not mis-use docking scores in their own research. This is an important fact that people who use docking in their studies should understand especially since docking has been so widely used as a cheap computational tool.
some grammar problems detected
Author Response

(The authors gave the same response as above.)

Reviewer 3 Report
The article deals with an important issue in the study of the activity of seed oil extracts with the help of solvents.
Environmental conditions, excessive consumption of synthetic drugs, food additives and preservatives have a significant impact on the biological production of free radicals, which underlie a number of diseases and pathological conditions, such as atherosclerosis, coronary heart disease, oncological diseases, etc. To correct these conditions and for prophylactic purposes, plant raw materials are increasingly used, containing a large set of antioxidants: vitamins, flavonoids and tannins, which have a mild effect on the body and relatively low toxicity. However, a controlled intake of antioxidants is necessary, since at a high content they become pro-antioxidants.
The practical use of antioxidants of plant origin for the regulation of free radical processes in the human body highlights the problem of quantifying the antioxidant effectiveness of complex preparations based on medicinal plant materials.
Based on the foregoing, the studies conducted by the authors are of interest to readers in the field under consideration.
However, the work has the following comments:
1. It is necessary to describe in more detail the methods considered in section “2. Materials and Methods". It is possible to generalize them in the form of a generalizing table highlighting the main advantages and disadvantages of the considered methods in comparison with other known methods.
2. It was necessary to characterize in more detail the methodology for conducting non-linear regression analysis (Section "2.7. Statistical analys"), as one of the most promising methods for studying the problem posed in the work.
3. Based on the data presented in Table 2, mathematical modeling could be performed to obtain regression equations containing key parameters that affect seed oil.
4. It would be necessary to give the predicted values of the percentage of inhibition depending on the concentration (Figure 3) and give the accuracy of the prediction. It would be interesting to apply forecasting methods using artificial neural networks, which, in particular, can be seen from the following work:
https://doi.org/10.3390/en15238919
5. According to Table 7, more detailed conclusions should be given, which will allow using the given data in carrying out similar studies.
6. In the conclusions, it was necessary to dwell in more detail on the approbation of the results obtained in practice in the treatment of various diseases. The introduction talks about various factors, such as diet, age, lifestyle, however, nothing is said about this in the conclusions.
7. Further prospects for work are not indicated, including with extracts of other seeds using various solvents in other countries, taking into account the specifics of climate, diseases, etc..
Author Response
We would like to thank the reviewers for spending their valuable time reviewing our manuscript. We highly appreciate their comments and suggestions. please see the attachment below for our responses to reviewers.

Round 2
Reviewer 3 Report
In general, the article has been finalized and can be published in my opinion.